# Using Variable Slope Total Derivative Estimations to Pick between and Improve Macro Models

Jonathan Leightner

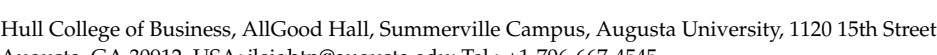

Hull College of Business, AllGood Hall, Summerville Campus, Augusta University, 1120 15th Street, Augusta, GA 30912, USA; jleightn@augusta.edu; Tel.: +1-706-667-4545

**Abstract:** Using the same data set, a researcher can obtain very different reduced form estimates just by assuming different macroeconomic models. Reiterative Truncated Projected Least Squares (RTPLS) or Variable Slope Generalized Least Squares (VSGLS) can be used to estimate total derivatives that are not model dependent. These estimates can be used to pick between competing macro models, improve current models, or create new models. A selected survey of RTPLS estimates in the literature reveals several common patterns: (1) as income inequality has surged around the world, the effect of changes in government spending (G), exports (X), and money supply (M-1) on Gross Domestic Product (GDP) have plummeted, (2) decreases in G, X, and M-1 cause GDP to fall more than equal increases in G, X, and M-1 cause GDP to rise, and (3) unusually large increases in G and M-1 cause their effect on GDP to plummet. These common patterns fit with a global glut of savings hypothesis, which predicts that an increase in savings will not cause an increase in production expanding investment. An appropriate model could be built around the idea that investors have a choice between investing to increase production or investing to earn rent or interest.

**Keywords:** omitted variables bias; total derivatives; choosing between macro models; global glut of savings; Keynesian model; government multipliers; money supply multipliers; export multipliers; production expanding investment; investment to own or rent

## 1. Introduction

Cogan et al. (2010) dramatically shows that a researcher can obtain very different estimates just by changing the macroeconomic model employed.[1] Specifically, Cogan et al. (2010) contrast the government spending multipliers derived from a Romer-Bernstein Keynesian model with a Taylor Keynesian model using the same data. They find that the Taylor model predicts a government spending multiplier of 1.4 in the first quarter, which then declines to zero in the 24th quarter of their data. In contrast the government spending multiplier estimated using the Romer-Bernstein Keynesian model is approximately one in the first quarter and steadily rises to 1.6 by the 8th quarter where it stays until the 16th quarter. They then find even a third set of results using a Smets-Wouters Keynesian model. Which model is closest to the truth, and upon which should we build government policy?

Often economists use models that are "hard wired" to produce the results that they want to find in order to "prove" the viewpoints that they want to prove. Too often economists gather around different campfires—one for classicalists, another for Keynesians, another for monetarists, one for rational expectations, another for real business cycle theorists, etc.—where we reinforce those around our own campfire with no accepted way of picking which campfire is best. What is needed are empirical tests that are not model dependent which can give guidance to model builders on what models would fit reality the closest.

Fortunately, in a recent *Journal of Risk and Financial Management* article, Leightner et al. (2021) explain how total derivatives between an independent and dependent variable can be estimated that capture the effects of omitted variables without having to model all the

possible ways that the two variables may interact. The key intuition that underlies their methods is that the combined influence of all omitted variables determines the relative vertical position of observations.[2] Thus, that relative vertical position can be used to capture the influence of omitted variables. Leightner, Inoue, and Lafaye de Micheaux's methods produce a separate slope estimate for every observation where the differences in these slope estimates are due to omitted variables. An important advantage of their methods is that they are not model dependent. For example, their methods can be used to estimate a total derivative between government spending (G) and gross domestic product (GDP) without having to assume either a classical model (that has built-in mechanisms to cause $d\text{GDP}/dG$ to equal zero) or a Keynesian model (that has built-in mechanisms to cause $d\text{GDP}/dG$ to be greater than one). Thus their methods can be used to estimate key total derivatives, which can then be used to pick between competing models or, even better, to create new models that are more realistic than any of our current choices.

Several points need to be emphasized. First, variable slope estimation methods are not a substitute for macroeconomic modelling; variable slope estimation and macroeconomic models are complements that, when used together, can help policy makers make better choices. One major disadvantage of variable slope estimation is that they cannot tell the mechanisms that underlie the estimated total derivatives; we need macroeconomic models to understand these mechanisms. Understanding the mechanisms that underlie the estimated total derivatives is crucial to the creation of optimal economic policies.

Second, the total derivative estimates produced by variable slope estimation methods capture all the ways that the independent and dependent variables are related. For example, if a government coordinates fiscal and monetary policy—cutting both spending and the money supply to combat inflation and increasing both to combat unemployment—then the total derivatives produced by variable slope estimation will capture that coordination. In other words, the variable slope fiscal spending multiplier estimate ($d\text{GDP}/dG$) does not hold the money supply constant if fiscal and monetary policy reinforce each other. Likewise, if a government increases spending to combat unemployment and that country's monetary authorities simultaneously cut the money supply out of fear of inflation, then the $d\text{GDP}/dG$ estimates will be diminished by the offsetting effects of contrary monetary policy.

Third, if government spending in year $t$ is correlated with government spending in year $t-1$ and government spending in $t-1$ continues to effect GDP in year $t$, then variable slope estimates of $d\text{GDP}/dG$ in year $t$ will include the lingering effects of government spending in $t-1$. This lingering effect can be diminished[3] by first differencing the data before conducting the analysis. A comparison of the $d\text{GDP}/dG$ estimates without first differencing to $d\text{GDP}/dG$ estimates after first differencing could give a researcher information about the strength of lingering effects. Furthermore, if the data is first differenced, then variable slope estimation could be used to estimate $d\text{GDP}_t/dG_t$, $d\text{GDP}_t/dG_{t-1}$, $d\text{GDP}_t/dG_{t-2}$, etc. An alternative way of gaining dynamic information from variable slope estimations is based upon changing the time period of analysis. For example, Leightner and Inoue (2008) explored the effects of China's exchange rate on the exchange rates of other Asian countries varying the time period of analysis from monthly to quarterly to annually.

Fourth, even without first differencing or adjusting the time period of analysis, variable slope estimates contain dynamic information because they produce a separate slope estimate for every observation. For example, East and West Germany were reunited in 1990 at which time German government spending rose noticeably. Leightner (2015) used a variable slope estimation method to estimate $d\text{GDP}/dG$ for Germany. He found that $d\text{GDP}/dG$ rose from 4.81 in the first quarter of 1988 to 5.19 in the fourth quarter of 1989 and then fell to 4.89 by the third quarter of 1992. The rise in $d\text{GDP}/dG$ between 1988 and 1989 could be due to rising private expectations leading up to the reunification, and the fall in $d\text{GDP}/dG$ fits what he found for many countries—unusually large increases in fiscal or monetary policy almost always result in a noticeable fall in the effectiveness of those policies.

This article will briefly explain the methods used by Leightner et al. (2021). The reason these methods are used is that they do not assume any model, and thus produce estimates that can be used to pick which model is best and to improve current models. One of these methods (reiterative truncated projected least squares—RTPLS) has been used to estimate many of the key relationships embedded in macroeconomic models. A selected survey of these estimates will be presented and their implications for macroeconomic modelling explained. This paper also presents evidence that there is a global glut of savings (where an increase in savings is not associated with an increase in production expanding investment) and challenges macroeconomic model builders to model this global glut of savings and its implications.

The remainder of this paper is structured as follows. Section 2 explains variable slope estimation methods and does not contribute significantly new information to the literature. Section 3 surveys (a) selected variable slope estimates in the literature and (b) published evidence that there is a global glut of savings. Section 3 contributes to the literature by (a) showing how the variable slope estimates in the literature fit with a global glut of savings, (b) making suggestions on how existing macroeconomic models can be improved based on these published variable slope estimates, and (c) suggesting an equation that could form the backbone of an entirely new macroeconomic model centered on the global glut of savings. Section 4 concludes.

## 2. Materials and Methods

If a researcher studying the yield of a wheat crop over several years did not include the amount and timing of rainfall, then his estimates would not be reliable. In this case the omitted variable—the amount and timing of rain—would affect the dependent variable (wheat yield) and the estimates of the relationships between wheat yield and some of the included independent variables like the amount of chemical fertilizer applied (the increased yield from applying chemical fertilizers is affected by the amount of water applied: Stone (2019) explains this and many other aspects of the green revolution). In this case, the researcher has an omitted variables problem that ruins all his statistics and estimates. However, if that same researcher did include the amount and timing of rain but excluded the number of times the farmer's cows got into the wheat field, then his estimates would not be biased. Cows like to eat wheat, but the number of times cows got into the wheat field would not affect the relationship between the dependent variable (wheat production) and the included independent variables (one of which is chemical fertilizer). In this case, the number of times cows got into the wheat field would just add "random" variation to the dependent variable (wheat yield) without affecting the estimates on how the included independent variables (chemical fertilizer, etc.) affect the dependent variable (wheat yield).

In other words, omitting independent variables from an estimation is only a "problem" if the omitted variable interacts with included independent variables—perhaps through a macroeconomic system of equations. Thus, if a researcher estimates Equation (1) while ignoring Equation (2), the resulting estimate of $\beta_1$ (how wheat yield is affected by fertilizer) is a constant when in truth $\beta_1$ varies with $q_i$ (the amount of water), and this ignoring of Equation (2) creates an omitted variables problem. The $\alpha$s and $\beta$s are coefficients to be estimated, Y is the dependent variable, X is the explanatory variable, $u$ is random error, and "$q_t$" represents the combined influence of all omitted variables plus any random variation in $\beta_1$ itself.

$$Y_t = \alpha_0 + \beta_1 X_t + u \tag{1}$$

$$\beta_1 = \alpha_1 + \alpha_2 q_t \tag{2}$$

One convenient way to model the omitted variable problem is to combine Equations (1) and (2) to produce Equation (3).

$$Y_t = \alpha_0 + \alpha_1 X_t + \alpha_2 X_t q_t + u_t. \tag{3}$$

The easiest way to explain variable slope estimation methods is to use a diagram like Figure 1. To construct Figure 1, I generated two series of random numbers, X and q, which ranged from 0 to 100 (in this example, I set $u_t = 0$). I then calculated the dependent variable Y as:

$$Y = 100 + 10\,X + 0.5\,qX \tag{4}$$

The true value for $dY/dX$ equals $10 + 0.5$ q. Since q ranges from 0 to 100, the true slope will range from 10 (when q = 0) to 60 (when q = 100). Thus q makes a 600 percent difference to the slope. In Figure 1, I identified each point with that observation's value for q. Notice that the upper edge of the data corresponds to relatively large qs—93, 97, 98, 98, 98, 98, and 99. The lower edge of the data corresponds to relatively small qs—2, 1, 2, 2, 7. This makes sense since as q increases so does Y, for any given X. For example, when X approximately equals 60, reading the values of q from top to bottom of Figure 1 produces 96, 88, 79, 65, 40, 28, and 17. Thus the relative vertical position of each observation is directly related to the values of q. If, instead of adding 0.5 qX in Equation (4), I had subtracted 0.5 qX, the smallest qs would be on the top and the largest qs on the bottom of Figure 1. Either way, the vertical position of observations captures the influence of q. In Figure 1, the true value for $dY/dX$ equals $10 + 0.5$ q; thus the slope, $dY/dX$, will be at its greatest numerical value along the upper edge of the data where q is largest and the slope will be at its smallest numerical value along the bottom edge of the data where q is smallest.

Now imagine that we do not know what q is and that we omit it from our analysis. In this case, OLS produces the following estimated equation: $Y = 44.314 + 37.412X$ with an R-Squared of 0.5415 and a standard error of the slope of 3.477. On the surface, this OLS regression looks successful, but it is not. Remember that the true equation is $Y = 100 + 10\,X + 0.5\,q\,X$. Since q ranges from 0 to 100, the true slope (true derivative) ranges from 10 to 60 and OLS produced a constant slope of 37.412. OLS did the best it could, given its assumption of a constant slope; OLS produced a slope estimate of approximately $10 + 0.5\,E(q) = 10 + 0.5(51.2) = 35.6$, which is reasonably close to the estimated 37.412 [E(q) = expected or mean value for q]. However, OLS is hopelessly biased by its assumption of a constant slope when, in truth, the slope is varying.

Although OLS is hopelessly biased when there are omitted variables that interact with the included variables, Figure 1 provides a very important insight—even when a researcher does not know what the omitted variables are, even when he or she have no clue how to model the omitted variables or measure them, and even when there are no proxies for the omitted variables, Figure 1 shows the researcher that the relative vertical position of each observation contains information about the combined influence of all omitted variables on the true slope.

For example, if there are 300 omitted variables that increase Y and 200 that decrease Y, the observations that are on the upper frontier will correspond to when the 300 omitted variables are at their highest levels and the 200 at their lowest.

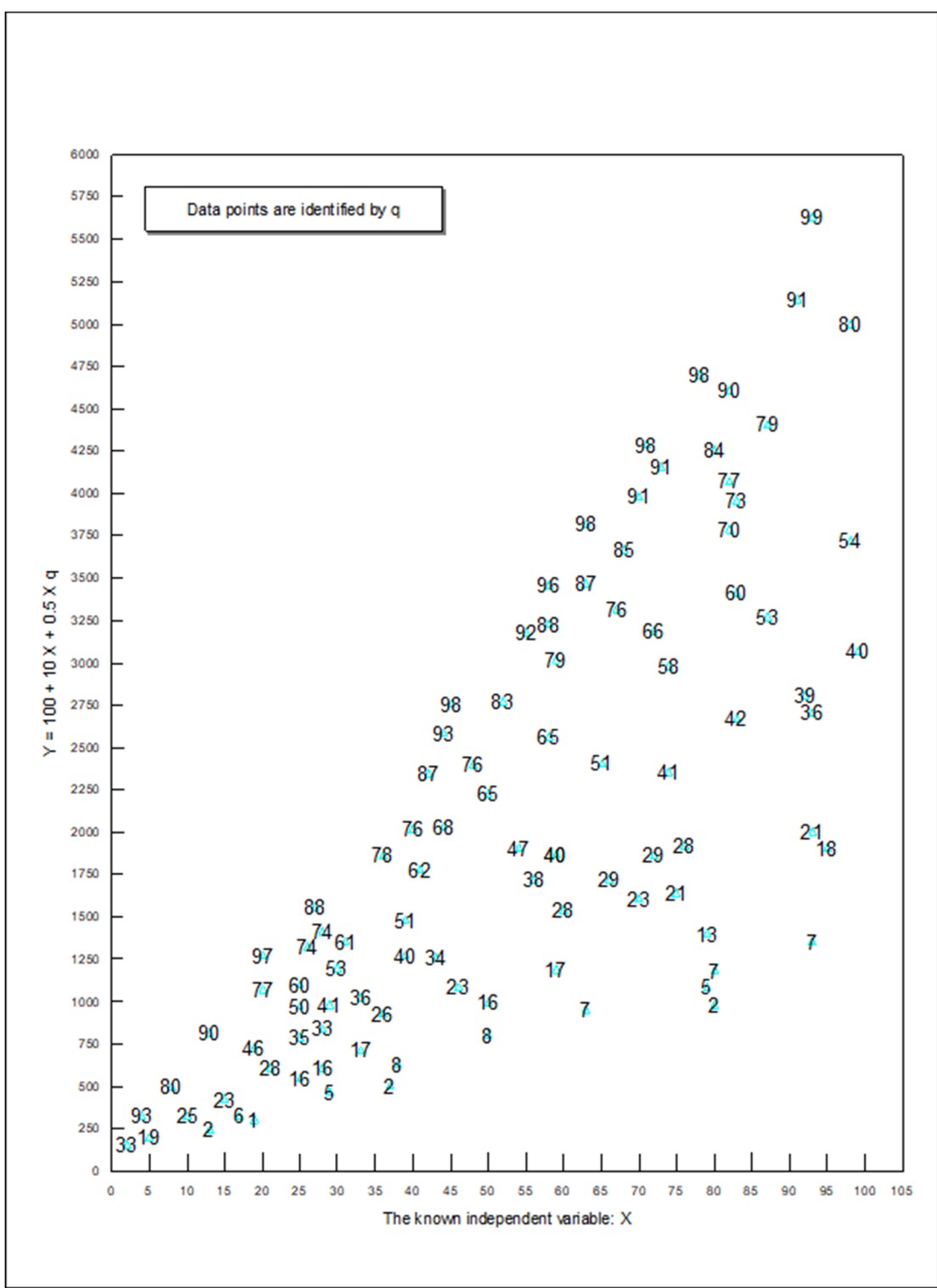

**Figure 1.** The intuition behind RTPLS.

How variable slope estimation methods exploit the relative vertical position of observations is based on Equation (9) which is derived from Equation (3) as follows.

$$(dY/dX)^{\text{True}} = \alpha_1 + \alpha_2 \, q_t \qquad \text{Derivative of Equation (3)} \qquad (5)$$

$$Y_t/X_t = \alpha_0/X_t + \alpha_1 + \alpha_2 \, q_t + u_t/X_t \qquad \text{Equation (3) divided by } X_t \qquad (6)$$

$$\alpha_1 + \alpha_2\, q_t = Y_t/X_t - \alpha_0/X_t - u_t/X_t \qquad \text{Equation (6) rearranged} \qquad (7)$$

$$(d\text{Y}/d\text{X})^{\text{True}} = Y_t/X_t - \alpha_0/X_t - u_t/X_t \qquad \text{From Equations (5) and (7)} \qquad (8)$$

Recall that $u_t$ is random error which should be relatively small, and $u_t/X_t$ even smaller. Leightner et al. (2021) show that eliminating $u_t/X_t$ from Equation (8) does not bias the results, and that elimination produces Equation (9).

$$d\text{Y}/d\text{X} = Y_t/X_t - \alpha_0/X_t \qquad (9)$$

Variable slope estimation methods produce an estimate of $\alpha_0$ which is then plugged into Equation (9) along with values for $Y_t$ and $X_t$ to produce a slope estimate for every observation. Variable Slope OLS (VSOLS) uses ordinary least squares to estimate Equation (1) and then plugs the resulting $\alpha_0$ into Equation (9). Variable Slope GLS (VSGLS) uses generalized least squares to estimate Equation (1) and then plugs the resulting $\alpha_0$ into Equation (9). Reiterative Truncated Projected Least Squares peels the data down layer by layer (like an onion) to produce slope estimates for every layer; each $Y_t/X_t$ is then subtracted from the corresponding layer's slope to produce a new dependent variable; and then a final regression is run between that new dependent variable and $1/X_t$ to find an $\alpha_0$ which is then plugged into Equation (9). The mathematical equations underlying RTPLS are explained in Leightner (2015).

Leightner et al. (2021) ran 5000 simulations each for the 27 combinations of the omitted variable making a 10 percent, 100 percent, and 1000 percent difference to the true slope, with random error being 0 percent, 1 percent, and 10 percent of the standard deviation of X, and with sample sizes of n = 100, 250, and 500. First, they found that VSGLS and RTPLS always noticeably outperformed VSOLS. This result did not surprise them since they had shown that Equation (3) implied heteroscedasticity, and Aitken (1935) showed that generalized least squares produces the best linear unbiased estimates (BLUE) under heteroscedasticity. Second, Leightner, Inoue, and Lafaye de Micheaux found that both VSGLS and RTPLS noticeably outperformed assuming that there are no omitted variables and using OLS except when random error effected the equation as much as the omitted variables affect it. This exception also makes sense since VSGLS and RTPLS use the relative vertical position of observations to capture the effects of omitted variables and relatively large amounts of random error would make it impossible to distinguish between the influence of omitted variables and randomness.

Third, Leightner, Inoue, and Lafaye de Micheaux found that when the effect of the omitted variables was ten times bigger than random error, using OLS while assuming there are no omitted variables produced approximately 3.8 times the error produced by VSGLS and RTPLS. Furthermore, when the effect of the omitted variables was one hundred times the size of random error, using OLS while ignoring omitted variables produced more than 27 times the error from using VSGLS or RTPLS. Fourth, when all error was due to omitted variables (random error = 0), then VSGLS produced more than 1.6 times the error of RTPLS. Since VSGLS has been shown to be BLUE, I believe that this result is due to RTPLS handling nonlinear effects better than VSGLS. In the most extreme case examined (omitted variables made 1000 percent difference to the true slope, zero random error, and n = 100) using OLS while ignoring the omitted variables problem produced 2138 times the error produced by RTPLS and VSGLS produced 34 times the error produced by RTPLS. However, RTPLS is more complicated and time consuming to conduct than VSGLS, and VSGLS can be performed using most standard regression programs (RTPLS cannot). Thus, the fact that RTPLS outperforms VSGLS when all error is due to omitted variables should not deter researchers from using VSGLS because VSGLS and RTPLS perform almost equally when there is random error.

VSGLS and RTPLS find total derivatives that show all the ways that the dependent and independent variables are related. Confidence intervals for variable slope estimates can be calculated using the central limit theorem.

$$\text{Confidence interval} = \text{mean} \pm (s/\sqrt{n})t_{n-1,\,\alpha/2} \qquad (10)$$

In Equation (10), "s" is the standard deviation, "n" is the number of observations, and $t_{n-1,\,\alpha/2}$ is taken off the standard t table for the desired level of confidence. Often a given estimate and the 2 estimates before and after it, and a 95% confidence level, are used to create a moving confidence interval (much like a moving average) for a given set of RTPLS estimates. This 95% confidence interval can be interpreted as meaning that there is only a five percent chance that the next RTPLS estimate will lie outside of this range if omitted variables maintain the same amount of variability that they recently have.[4]

### 3. Results

Many RTPLS estimates published to date provide evidence for a global glut of savings. However, before I survey those RTPLS estimates, let us think about how modeling a global glut of savings would affect macroeconomic models. There is only one fundamental difference between the equations that underlie a classical and a Keynesian economic model—the classical model includes an equation that equates labor supply and labor demanded and the Keynesian model omits that equation. The inclusion or exclusion of a perfectly functioning labor market equation changes the fundamental functioning of the macroeconomic model. In the classical model, aggregate supply is vertical at full employment; in the Keynesian model aggregate supply is not vertical. In the classical model aggregate demand is only shifted by the money supply; in the Keynesian model aggregate demand is shifted by many things including government spending, taxes, consumer expectations, private investment, and the money supply. In a classical model, changes in government spending (unless financed by taxes) will not change employment or GDP; in a Keynesian model, changes in government spending will change employment, GDP, and inflation. In a classical model, increasing the money supply will cause inflation and not help employment or GDP; in a Keynesian model, increasing the money supply can help GDP and employment. Classical models imply that the government should not try to help the economy; Keynesian models imply that the government should try to help the economy (Froyen 2013). All of the above differences are due to the inclusion or exclusion of only one equation—an equation that equates labor supply and labor demanded.

If the assumption of equilibrium or disequilibrium in the labor market can so fundamentally change the functioning of a macroeconomic model, then how would the possibility of a glut of savings (Bernanke 2005) change the functioning of our models? Different scholars define "global glut of savings" different ways—Bernanke (2005) focused on it as a primary cause of the US trade deficit. Our simplest growth models are built on the idea that savings constrains growth producing investment, and I am defining a "global glut of savings" as when savings increases but production expanding investment does not increase.

Our simplest growth models imply that if we increase taxes on the poor and cut taxes to the rich, then savings will increase which will increase investment which then would increase growth. Contrary to this "anti-equality" argument, Leightner (2015) argues that growth producing invest requires two things—savings to fund the investment and the belief that what the investment would produce will sell. If investors do not see evidence of sufficient consumption to justify their investment, then the investors will not invest no matter how much excess savings exists and no matter how low interest rates fall. Thus Leightner (2015) proposes an alternative pro-equality argument—cutting taxes to the poor and increasing taxes on the rich will increase consumption, which will stimulate investment, which then would increase growth.

Leightner (2015) admits that there will be times that insufficient savings constrains investment and other times that consumption drives investment. However, Leightner (2015) believes that the current world is suffering from a glut of savings, and insufficient

consumption. He found evidence of excess savings from three groups—corporations, nations, and banks, and I will now add households to that list.

In the fourth quarter of 2009, US corporations held US\$ 74.7 billion in cash (not including short term investments) on hand or in US banks; by the fourth quarter of 2021 these US corporate cash holdings had grown 3.4 fold to US\$ 257.7 billion (U.S. Census Bureau 2022). Globally, corporate cash holdings plus short-term investments in the second quarter of 2021 were "at an all-time high of US\$ 6.84 trillion ... That is 45% higher than the average in the five years preceding the pandemic" (Hirtenstein 2021). Leightner (2015, p. 3) summarizes his survey of what corporations around the world are doing with their excess savings by saying,

> It is as if corporations are willing to sit on their cash, use their cash to buy back their own stock, or use their cash to pay greater dividends; in other words, they are willing to do anything with their cash except invest it in expanding production.

One way that nations save is by accumulating foreign reserves. According to the International Monetary Fund (2022) in the fourth quarter of 2021, total global foreign exchange reserves were 12.9 trillion US dollars (of which 7.1 trillion are claims in US dollars and 2.5 trillion are claims in euros). World holdings of foreign reserves rose dramatically after the 1997 Asian financial crisis.

The Office of the Comptroller of the Currency (2022) reports that the loan-to-deposit ratio at US banks dropped "from about 81 percent at year-end 2019 to 57 percent in mid-2021." The difference between 81 percent and 57 percent represents "about US\$ 3 trillion in 'excess' deposits—equal to roughly 17 percent of total deposits." This report also documented that the personal savings rate of US households was approximately 3 percent immediately prior to 2007, increased to 10 percent on average from 2007 to 2012, fell some after 2012, but rose during the pandemic and continues to be 10.3 percent in January 2022. The January 2022 US household savings rate of 10.3 percent is noticeably higher than the average savings rate of 7.2 percent for 2010 to 2019.

Mian et al. (2021) show that "there has been a large rise in savings by Americans in the top 1% of the income or wealth distribution over the past 40 years." They also argue that this "saving glut of the rich" has financed the dissavings by the non-rich and the government (through the rich acquiring money market funds, time deposits, bond purchases, mutual funds, and defined contribution pensions). They emphasize that this savings glut has "not boosted investment.

> The findings also call into question the idea in many macroeconomic models that a rise in savings automatically translates into additional capital formation. In the United States over the past 40 years, the substantial rise in savings by the top 1% has been associated with dissaving by the government and the bottom 90%, as investment actually fell." (Mian et al. 2021, p. 4)

Moreover, the fraction of world gross domestic product (GDP) devoted to investment *declined* over the past 40 years (Barsky and Easton 2021, p. 3). Thus, while savings by corporations, banks, nations, and rich households have surged, investment in expanding production has fallen. Clearly savings does not constrain production expanding investment; there is a global glut of savings.

I have not seen in the literature a way to incorporate a glut of savings into a functioning macroeconomic model, and it is beyond the scope of this paper to produce an entire macro model built on a glut of savings and to explore its implications. However, I have some ideas for the key equation (Equation (11)) upon which such a model could be built. Equation (11) can be derived from the GDP expenditures equation where $I_p$ = investment in tools, buildings, and equipment (i.e., production expanding investment), $I_o$ = investment to earn rent or interest from owning, U = undesired changes in inventories, S = private

savings, T = taxes, G = government purchases of final goods and services, M = imports, and X equals exports.

$$I_p + I_o + U = S + T - G + M - X \tag{11}$$

At equilibrium undesired changes in inventories (U) would equal zero, and $I_p$ (production expanding investment) + $I_o$ (investment to own) is constrained to equal the sum of private savings (S) plus government savings (T − G) plus foreign savings (M − X). Equation (11) could form the backbone of a glut of savings macro model. However, additional equations in such a model would need to model the forces that would cause investors to shift between production expanding investment ($I_p$) and investment to earn rent or interest ($I_o$), where production expanding investment ($I_p$) would be positively related to consumption and Investment to own ($I_o$) positively related to interest rates.

If there is insufficient consumption to justify production expanding investment ($I_p$), then the rich invest in money market funds, time deposits, bond purchases, mutual funds, and defined contribution pensions, as described by Mian et al. (2021); in other words, the rich shift to investing in "owning" ($I_o$) instead of in "producing more" ($I_p$). This shift would be fine if the money invested in these types of accounts was then lent out to those who used the money to purchase tools, equipment, and buildings (expand production); however, as shown above, the loan to deposit ratio at banks is plummeting.

Furthermore, this shift from investing in expanding production to investing in owning can create a self-reinforcing downward spiral. For example, if the rich purchase more houses that they intend to rent, the price of houses increase, making it harder for the poor to purchase a house. The poor are then forced to rent. This process can lead to income being redistributed from the poor to the rich, causing consumption to fall, making the return from production expanding investment to fall, encouraging more rich to purchase more houses, etc. Such a phenomenon is well documented for China.

It is interesting to consider how modelling a glut of savings would affect our macroeconomic models. In important ways, a glut of savings would cause a macroeconomic model to function like a Keynesian model. In a Keynesian model that allowed income distributional effects, shifting income from the rich to the poor would increase the marginal propensity to consume (MPC) and consumption. Since the government spending multiplier in our simplest Keynesian models is 1/(1-MPC), a redistribution from the rich to the poor would increase MPC and the multiplier effect of both fiscal and monetary policies. Recall that a global glut of savings implies that there is insufficient consumption to justify production expanding investment. Thus, in a global glut of savings macro model, redistributing income from the rich to the poor would increase consumption which would give a reason to invest in production expanding ways. Thus, both a Keynesian model and a global glut of savings model would advocate driving growth by taking from the rich to give to the poor.

The first Keynesian models just assumed that there are wage rigidities that prevent equilibrium in the labor market. Neoclassical economists criticized these early Keynesian models for not being based on "rational' decisions. New Keynesian economics focuses on providing a reason for disequilibrium in the labor market based on rational choices (Froyen 2013). These "rational" bases include menu costs, insider/outsider issues, producer gains from paying more than the wage paid by other firms (efficiency wages), market power in output markets, positive costs of forming expectations, heterogeneous expectation formation, etc. (Froyen 2013; Jump and Levine 2019; Gabaix 2020). Equation (11) above provides another "rational" reason for a model that would have many of the characteristics and predictions of Keynesian and New Keynesian models. The rational choice underlying Equation (11) is the choice between investing in ways that would expand production versus investing to earn rent or interest, where the later choice can produce a global glut of savings.

Several RTPLS estimates provide evidence for the global glut of savings hypothesis and its implications. First Leightner (2015) documents that income inequality is increasing around the world which would cause the marginal propensity to consume (MPC) to fall, which in turn would cause both fiscal and monetary multipliers to decline. His RTPLS estimates show that the government spending multiplier ($d$GDP/$d$G) had recently fallen for

Japan, the United Kingdom, the USA, Brazil, the Russian Federation and for the 17 countries that use the Euro. The average fall in $d$GDP/$d$G for these countries was from 5.621 to 4.530 for an average fall of 18.6 percent.

When interpreting these multipliers, it is essential to remember that they are reduced form total derivatives that show all the ways that GDP and government spending are correlated. Thus, if a given country coordinated fiscal and monetary policy to combat recessions, then the RTPLS estimates of $d$GDP/$d$G would capture the combined effects of both fiscal and monetary policy. Likewise, if the government spent money on constructing new roads and private investment occurred along those new roads, then the RTPLS estimates of $d$GDP/$d$G would capture the effects of building the roads and the stimulated private investment (and also capture any crowding out of private investment due to the interest rate rising, etc.).

Leightner (2015) found that the money supply multiplier ($d$GDP/$d$Money) had also recently fallen for Japan, the United Kingdom, the USA, Brazil, and the Russian Federation from an average of 7.397 to 3.994 for an average fall of 51.3 percent (this multiplier could not be estimated for the euro using countries because they do not have independent control over their money supplies). For all countries with data on M-1, Leightner (2015) measured the money supply as M-1; however, a lack of M-1 data forced Leightner (2015) to use M-2 data for the Russian Federation and M-0 data for the UK. Furthermore Leightner's (2015) RTPLS estimates show that the export (X) multiplier ($d$GDP/$d$X) had recently fallen for these same countries and for the 17 countries that use the Euro. The average fall in $d$GDP/$d$X was from 4.211 to 2.314 for an average fall of 33.9 percent. Leightner (2015) concludes that the recent increase in inequality has caused government spending, money supply and export multipliers to dramatically fall, just as Keynesian and global glut of savings models would predict.

Leightner and Inoue (2014) showed how Thailand's fiscal, monetary, and trade multipliers tended to increase or stabilize under governments with pro-poor policies (initiated by the Thai-Love-Thai and People's Power political parties) and fall when governments emphasized maintaining the status quote or helping the rich and middle classes (policies associated with the Democratic Party of Thailand). The highest government spending multipliers that Leightner (2015) found were for Japan while the ratio of the wage bill to profits increased from 0.902 to 3.065 between 1952 and 1981. However, when the Japanese government ended its pro-equality policies by early 1980, Japan's $d$GDP/$d$G fell from 9.92 to 6.79 for a 31.5 percent decline.

Leightner and Haiqi (2016) use RTPLS to estimate the change in GDP due to a change in different types of taxes: $d$GDP/$d$(Property Tax), $d$GDP/$d$(Corporate Tax), $d$G/$d$(Income Tax) and $d$GDP/$d$(Sales Tax). Again their RTPLS estimates are reduced form estimates that capture all the ways that taxes and GDP are related including through the spending of those tax revenues. They used annual data from 1970 to 2012 for 23 countries where GDP was measured in millions of US dollars and each tax was measured as tax revenue as a percent of GDP. For 13 of the 23 countries examined after 2008, they found that property and corporate taxes are the best taxes to increase (increases GDP the most) and individual income taxes and sales taxes are the worse to raise. They conclude that shifting the tax burden from the poorest (who are most affected by income and sales taxes) to the richest (who are most affected by corporate and property taxes) would increase growth for those countries. This conclusion fits with Keynesian and a global glut of savings models. The above surveyed RTPLS results imply that we should include in our macroeconomic models equations that show how income distribution affects the marginal propensity to consume, and, thus, government spending, money supply, and export multipliers.

Leightner (2015) found that every time the US Federal Reserve system (FED) increased the money supply, $d$GDP/$d$Money fell, and every time the FED decreased the money supply, $d$GDP/$d$Money rose. For example, he found that the negative effects of the FED decreasing the money supply in 1997 was 24 percent larger than the positive effects of

increasing the money supply in 1994. This result has major implications for the FED's current plans to decrease the money supply in 2022 to combat inflation.

Furthermore, Leightner (2015) found that every time governments increased government spending by unusually large amounts, $d$GDP/$d$G fell noticeably (for examples, Greece in Q4 of 2009, France in Q1 of 1977 and 1979, and the Netherlands in Q1 of 2006). For both the USA and the UK, $d$GDP/$d$G began falling when the trajectory of future increases in government spending increased and rebounded some when government spending stabilized at a more consistent level between consecutive years. Moreover, Leightner (2012) found that the countries that increased spending the most to offset the Great Recession of 2008–2009 tended to be the countries that had suffered declining $d$GDP/$d$G for several years—Australia, Indonesia, and Sri Lanka—and their big spending in 2008–2009 was correlated with $d$GDP/$d$G falling even more. In contrast, countries that increased government spending moderate amounts between 2008 and 2009 in response to the Great Recession did not suffer declining $d$GDP/$d$G. In the Russian Federation (2000–2008), Austria (1977–1995), and Finland (1974–1991) government spending rose noticeably in the first quarter of every year and then increased less rapidly for the rest of the year; meanwhile $d$GDP/$d$G fell in the first quarter, then rose for the remainder of the year.

A global glut of savings model can be used to explain why increases in government spending, the money supply, and exports have a smaller multiplier effect than equal declines in them. Anything that would make the glut of savings worse (increase the right-hand side of Equation (11)) would have a stronger multiplier effect than things that would lessen the glut of savings. Increases in government spending (G) would lessen the right-hand side of Equation (11), reducing the savings glut and, thus, have a smaller multiplier effect than decreases. Decreases in government spending would worsen the savings glut, and, thus, have a stronger multiplier effect than increases. The analysis of exports (X) parallels that of government spending.

Increases in the money supply would cause interest rates to fall, causing a shift from $I_o$ to $I_p$ (a fall in the interest rate reduces the return from owning accounts that produce interest) using up part of the savings glut. However, decreases in the money supply would drive up interest rates causing a shift from $I_p$ to $I_o$, worsening the glut of savings. Thus decreases in the money supply would have a stronger multiplier effect than increases in the money supply.

## 4. Conclusions

Cogan et al. (2010) show that what theoretical model a researcher employs can dramatically affect the estimates he or she produces. One way to pick the model to use is to justify it on theoretical bases. However, macro economists do not agree on the appropriate theoretical model. For example, a Classical model would predict that the government spending multiplier ($d$GDP/$d$G) would equal zero, if the spending was not financed by increased taxes. However, deleting one equation from the Classical model—the equation that guarantees equilibrium in the labor market—changes the Classical model into a Keynesian model where $d$GDP/$d$G is noticeably greater than one. Moreover, differences in macro models extend far beyond just a Classical versus Keynesian distinction. Indeed Cogan et al. (2010) shows that dramatically different $d$GDP/$d$G multipliers are estimated under an old Keynesian model versus a new Keynesian model in spite of using the same set of data.

What macro economists need is a way to estimate total derivatives that are not model dependent and that are not biased by the omitted variables problem. Two such methods are Reiterative Truncated Projected Least Squares (RTPLS) and Variable Slope Generalized Least Squares (VSGLS). RTPLS and VSGLS use the relative vertical position of observations to capture the combined influence of all omitted variables. RTPLS and VSGLS produce a separate slope estimate for every observation where differences in these slope estimates are due to omitted variables. One major advantage of RTPS and VSGLS estimates is that you can see how the relationship between the independent variable and the dependent

variables have changed over time due to omitted variables. Another major advantage is data is only needed on the independent and dependent variables—a researcher does not have to develop an entire macroeconomic model and estimate every equation in that model in order to derive a total derivative. However, one disadvantage of RTPLS and VSGLS is their estimates do not tell the researcher the mechanisms through with the estimated relationship functions. RTPLS and VSGLS estimates can be used to pick which of competing models best fits reality.

The vast majority of RTPLS estimates in the literature support a Keynesian model over a Classical model in that the government spending multipliers, money supply multipliers, and export multipliers all are noticeably greater than one. However, these RTPLS estimates do not support a simple Keynesian model where these multipliers are constant throughout time. Instead, these estimates show that (1) as income has become less equal around the world, these multipliers have plummeted, (2) many times increases in government spending, the money supply, and exports cause GDP to rise less than decreases cause GDP to fall. Furthermore, unusually large increases in government spending and the money supply are associated with dramatic falls in their multiplier effects.

Possible ways to adjust our current macroeconomic models to address these RTPLS results include (1) set a dummy variable to zero when government spending increases and to one when government spending decreases in order to capture the asymmetric effects of increases versus decreases in government spending, (2) also use dummy variables for increases versus decreases in the money supply and exports, (3) if there is enough observations to do so, dummy unusually large increases in government spending and the money supply, and (4) model how income distribution affects the marginal propensity to consume.

However, better than just adjusting our current macroeconomic models would be creating a new model built upon a glut of savings. Such a model could be constructed around that idea that investors have a choice between investing in ways to expand production ($I_p$) or investing in order to earn interest or rent ($I_o$). An increase in consumption would drive investment to expand production ($I_p$) while an increase in interest rates would drive investment to own ($I_o$). Such a model would fit with the RTPLS estimates in the current literature. The RTPLS estimates surveyed here imply that forces that worsen the global glut of savings (decreases in government spending and in the money supply) have stronger multiplier effect than forces that reduce the global glut of savings (increases in government spending and in the money supply).

**Funding:** This research received no external funding.

**Data Availability Statement:** Not applicable.

**Conflicts of Interest:** The author declares no conflict of interest.

## Notes

[1] Cogan et al. (2010) provide the clearest and most dramatic evidence that picking a model can determine the empirical results produced. Their article was seminal—as of June 2022 it has been cited 938 times. I know of no recent publications that reproduce their results nor do I know of papers that repudiate their results. It inspired this paper.

[2] This intuition was first published in (Branson and Lovell 2000).

[3] One fruitful avenue for future research would include using simulation tests to see to what degree first differencing diminishes this lingering effect. Perhaps, first differenceing would totally eliminate it. To date simulation tests have not been conducted on any of the methods suggested in this paragraph.

[4] Published applications of RTPLS can be found in *Journal of Risk and Financial Management, Biomedical Journal of Scientific & Technical Research, International Journal of Contemporary Mathematical Sciences, Economics Bulletin, European Journal of Operations Research, Journal of Central Banking Theory and Practice, International Journal of Financial Research, Economies, China Economic Policy Review, Applied Economic Letters, Frontiers of Economics in China, China & World Economy, Pacific Economic Review, The Japanese Economy: Translations and Studies, Journal of Productivity Analysis, Economy, International Economics & Finance Journal, Advances in Decision Sciences, International Journal of Economic Issues, Global Economy Journal, Journal of Financial Economic Policy,* and *Contemporary Social Science.*

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
