# Peer review of "Using Variable Slope Total Derivative Estimations to Pick between and Improve Macro Models"

_jrfm, doi:10.3390/jrfm15060267_

Round 1

Reviewer 1 Report

This is an excellent paper. The analysis is very tight and represents a novel, non-trivial contribution the the literature. The JRFM is a natural outlet for this type of studies, and I am in favor of publication, basically as-is. I have two main questions in reaction to the paper, neither of which undermines the analysis as far as it goes.

First, since the models addressed in the paper are static in nature, it is impossible to asses relevant questions related to the estimation of e.g. short- and long-run elasticities. This is quite a tough task and would likely result in a very different paper. Nonetheless, I think that the author should discuss this issue in the introduction as well as fruitful avenues for future research in the concluding remarks; this would most likely increase the appeal of the paper.

Second, while the underlying methods are rather techincal, they are borrowed from previous work from the same author (with co-authors); the introduction appears not to properly deliver to the reader how this paper's findings enrich our understanding of the modeling issues in the macroeconomic realm. I would suggest that the author makes a bigger effort in highlighting the differences (with respect to modelling approaches, to divergence in empirical findings based on artifical data sets, etc) between the body of the reference literature and the current paper. It is indeed crucial to make clear at the outset, and in a non-technical manner, how the results reported in the present paper contribute to our understanding of model selection and validation in a structural setting. My general advice here is intended to facilitate reading and render the paper more appealing in terms of the interpretation of the results.

Finally, the paper would greatly benefit from some more polishing of its exposition and proof-reading. E.g., line 16 ("increase" to be fixed into "increases", twice on the same line); line 65 ("compliments" to be fixed into  "complements"); line 72 ("imbedded" not so common spelling nowadays, better use "embedded"); lines 214 to 223 belong to a footnote and should not appear in the main text.

Author Response

Referee One

This is an excellent paper. The analysis is very tight and represents a novel, non-trivial contribution the the literature. The JRFM is a natural outlet for this type of studies, and I am in favor of publication, basically as-is. I have two main questions in reaction to the paper, neither of which undermines the analysis as far as it goes.

Thank you.

First, since the models addressed in the paper are static in nature, it is impossible to asses relevant questions related to the estimation of e.g. short- and long-run elasticities. This is quite a tough task and would likely result in a very different paper. Nonetheless, I think that the author should discuss this issue in the introduction as well as fruitful avenues for future research in the concluding remarks; this would most likely increase the appeal of the paper.

To the introduction I have added:

Second, the total derivative estimates produced by variable slope estimation methods capture all the ways that the independent and dependent variables are related.  For example, if a government coordinates fiscal and monetary policy – cutting both spending and the money supply to combat inflation and increasing both to combat unemployment – then the total derivatives produced by variable slope estimation will capture that coordination. In other words, the variable slope fiscal spending multiplier estimate (dGDP/dG) does not hold the money supply constant if fiscal and monetary policy reinforce each other. Likewise, if a government increases spending to combat unemployment and that country’s monetary authorities simultaneously cut the money supply out of fear of inflation, then the dGDP/dG estimates will be diminished by the offsetting effects of contrary monetary policy. 

               Third, if government spending in year t is correlated with government spending in year t-1 and government spending in t-1 continues to effect GDP in year t, then variable slope estimates of dGDP/dG in year t will include the lingering effects of government spending in t-1.  This lingering effect can be diminished by first differencing the data before conducting the analysis. A comparison of the dGDP/dG estimates without first differencing to dGDP/dG estimates after first differencing could give a researcher information about the strength of lingering effects. Furthermore, if the data is first differenced, then variable slope estimation could be used to estimate dGDPt/dGt, dGDPt/dGt-1, dGDPt/dGt-2, etc. An alternative way of gaining dynamic information from variable slope estimations is based upon changing the time period of analysis. For example, Leightner and Inoue (2008) explored the effects of China’s exchange rate on the exchange rates of other Asian countries varying the time period of analysis from monthly to quarterly to annually.

               Fourth, even without first differencing or adjusting the time period of analysis, variable slope estimates contain dynamic information because they produce a separate slope estimate for every observation. For example, East and West Germany were reunited in 1990 at which time German government spending rose noticeably. Leightner (2015) used a variable slope estimation method to estimate dGDP/dG for Germany.  He found that dGDP/dG rose from 4.81 in the first quarter of 1988 to 5.19 in the fourth quarter of 1989 and then fell to 4.89 by the third quarter of 1992. The rise in dGDP/dG between 1988 and 1989 could be due to rising private expectations leading up to the reunification, and the fall in dGDP/dG fits what he found for many countries – unusually large increases in fiscal or monetary policy almost always result in a noticeable fall in the effectiveness of those policies.

               And in the conclusion, I write:

Possible ways to adjust our current macroeconomic models to address these RTPLS results include (1) set a dummy variable to zero when government spending increases and to one when government spending decreases in order to capture the asymmetric effects of increases versus decreases in government spending, (2) also use dummy variables for increases versus decreases in the money supply and exports, (3) if there is enough observations to do so, dummy unusually large increases in government spending and the money supply, and (4) model how income distribution affects the marginal propensity to consume.

However, better than just adjusting our current macroeconomic models would be creating a new model built upon a glut of savings.  Such a model could be constructed around that idea that investors have a choice between investing in ways to expand production (Ip) or investing in order to earn interest or rent (Io).  An increase in consumption would drive investment to expand production (Ip) while an increase in interest rates would drive investment to own (Io).  Such a model would fit with the RTPLS estimates in the current literature. The RTPLS estimates surveyed here imply that forces that worsen the global glut of savings (decreases in government spending and in the money supply) have stronger multiplier effect than forces that reduce the global glut of savings (increases in government spending and in the money supply).

Finally, I have added footnote 2: One fruitful avenue for future research would include using simulation tests to see to what degree first differencing diminishes this lingering effect.  Perhaps, first differenceing would totally eliminate it. To date simulation tests have not been conducted on any of the methods suggested in this paragraph.

Second, while the underlying methods are rather techincal, they are borrowed from previous work from the same author (with co-authors); the introduction appears not to properly deliver to the reader how this paper's findings enrich our understanding of the modeling issues in the macroeconomic realm. I would suggest that the author makes a bigger effort in highlighting the differences (with respect to modelling approaches, to divergence in empirical findings based on artifical data sets, etc) between the body of the reference literature and the current paper. It is indeed crucial to make clear at the outset, and in a non-technical manner, how the results reported in the present paper contribute to our understanding of model selection and validation in a structural setting. My general advice here is intended to facilitate reading and render the paper more appealing in terms of the interpretation of the results.

            I have added the following paragraph to the introduction.

The remainder of this paper is structured as follows.  Section 2 explains variable slope estimation methods and does not contribute significantly new information to the literature. Section 3 surveys (a) selected variable slope estimates in the literature and (b) published evidence that there is a global glut of savings.  Section 3 contributes to the literature by (a) showing how the variable slope estimates in the literature fit with a global glut of savings, (b) making suggestions on how existing macroeconomic models can be improved based on these published variable slope estimates, and (c) suggesting an equation that could form the backbone of an entirely new macroeconomic model centered on the global glut of savings.  Section 4 concludes. 

Finally, the paper would greatly benefit from some more polishing of its exposition and proof-reading. E.g., line 16 ("increase" to be fixed into "increases", twice on the same line); line 65 ("compliments" to be fixed into  "complements"); line 72 ("imbedded" not so common spelling nowadays, better use "embedded"); lines 214 to 223 belong to a footnote and should not appear in the main text.

It is disturbing how I can proofread a paper many times and not see mistakes like the ones you point out, because I know what I intended to write and I read what I intended.  Thank you for pointing these mistakes out.  I have changed the paper accordingly.

Reviewer 2 Report

The paper is interesting however more work is needed before it get accepted. I suggest the following:

1) The authors discuss about models in general, however they don't really analyze New Keynesian models.

2) Thus, the discussion and analysis is a bit simplified with many of the current reference models missing.

3) An application discussing all contemporary models used and analyzed from the perspective of this approach would be welcome.

Author Response

The paper is interesting however more work is needed before it get accepted. I suggest the following:

1) The authors discuss about models in general, however they don't really analyze New Keynesian models. 

            I have added the following to the results section:

The first Keynesian models just assumed that there are wage rigidities that prevent equilibrium in the labor market.  Neoclassical economists criticized these early Keynesian models for not being based on “rational’ decisions.  New Keynesian economics focuses on providing a reason for disequilibrium in the labor market based on rational choices (Froyen, 2013). These “rational” bases include menu costs, insider/outsider issues, producer gains from paying more than the wage paid by other firms (efficiency wages), market power in output markets, positive costs of forming expectations, heterogeneous expectation formation, ect. (Froyen, 2013, Jump and Levine, 2019, and Gabaux, 2020). Equation 11 above provides another “rational” reason for a model that would have many of the characteristics and predictions of Keynesian and New Keynesian models. The rational choice underlying equation 11 is the choice between investing in ways that would expand production versus investing to earn rent or interest, where the later choice can produce a global glut of savings.

A literature survey that details all the contributions of all the authors that have taken a New Keynesian approach would constitute an entire paper by itself.

2) Thus, the discussion and analysis is a bit simplified with many of the current reference models missing.

            A discussion and analysis of all the current models would fill an entire book. 

3) An application discussing all contemporary models used and analyzed from the perspective of this approach would be welcome.

I was given 10 days to make this revision.  Adding an application and applying all contemporary models to that application would take me an entire year and the resulting manuscript would be book length.  However, I have added a list of some open access applications of RTPLS.  At the end of the second section I write:

Open access articles that contain RTPLS estimates include Leightner (2021) and Leightner, Inoue, and Lafaye de Micheaux (2019) on the spread rate of Covid-19, Leightner (2020) on the Philip’s Curve, Leightner (2018) on the effects of China’s reserves on China’s and the USA’s exports, and Leightner and Inoue (2014) on dGDP/dG, dGDP/d(Monetary Base), dGDP/dExports, dGDP/d(exchange rate), and dGDP/d(foreign reserves) for Thailand.

Reviewer 3 Report

Dear Authors,

I am very glad to inform you that I have got scope to review a paper entitled Using Variable Slope Total Derivative Estimations to Pick Between and Improve Macro Models. Though you have followed the scientific methods to write your paper, there are shortcomings. However, some major revisions are required before publishing it. There are given below: 

1. Abstract:  First of all, according to journal’s guidelines, the abstract should be structured and emphasize your work’s aim first and better. Moreover, the theoretical and practical implications of your work should be very clear when reading the Abstract, but I cannot revise them at the moment.

2. Introduction: Secondly, I revise the same issues in the Introduction, which will need great work to be improved. In particular, I suggest adopting a commonly used scheme, structuring this section as follows:

•       Broad theme or topic;
•       Academic and practical importance;
•       Literature summary;
•       Gaps, inconsistencies, controversies to be addressed;
•       Focus of the study and research question(s), here or in Section 2;
•       Applied methods;
•       Main results and contributions;
•       Structure of the article (remainder).
At the moment, some of these parts are a little bit vague (i.e., theoretical and practical relevance, research gap) or missing (i.e., paper’s main results and implications, remainder). Moreover, you should try to better position your paper into the current scientific debate. In this sense, take particular care also to avoid supporting recent trends with old references. This happens even in the first lines of the article and several other times throughout the paper (i.e., also with references published in 2010). I also suggest providing a brief explanation of what is variable slope and how developed the macro model since the Introduction to help international readers contextualizing your analysis and generalizing its results.

3. Please add a literature review part to identify the methods and variables. Theoretical discussion and later implications of theory in research are very crucial. Furthermore, I am not biassed in your introduction. , considering Section 2, I suggest mobilising part of those concepts in the other paragraphs of the literature review since I cannot find any great value added by this section of the introduction.  

4. Please check the equations of Methods part. I need justification more as why did you choose the method. 

5. Result part is ok.

6. Fifth, I suggest also giving more attention to developing the conclusion of your results, trying to better link it to the current scientific debate and the literature leveraged in Section 2 and 3. Reading this paragraph, I still cannot capture the distinctive elements that support the relevance of your study. Similarly, the theoretical and practical implications of your work, as well as its limitations and possible further developments, should be added in the Conclusion paragraph or in a separate section of the paper. Some great work is required in this regard.

Thank you 

Good Luck with your research.

Author Response

Referee Three

Dear Authors,

I am very glad to inform you that I have got scope to review a paper entitled Using Variable Slope Total Derivative Estimations to Pick Between and Improve Macro Models. Though you have followed the scientific methods to write your paper, there are shortcomings. However, some major revisions are required before publishing it. There are given below: 

  1. Abstract:  First of all, according to journal’s guidelines, the abstract should be structured and emphasize your work’s aim first and better. Moreover, the theoretical and practical implications of your work should be very clear when reading the Abstract, but I cannot revise them at the moment.

The journal’s guidelines say:

  • Abstract:The abstract should be a total of about 200 words maximum. The abstract should be a single paragraph and should follow the style of structured abstracts, but without headings: 1) Background: Place the question addressed in a broad context and highlight the purpose of the study; 2) Methods: Describe briefly the main methods or treatments applied. Include any relevant preregistration numbers, and species and strains of any animals used. 3) Results: Summarize the article's main findings; and 4) Conclusion: Indicate the main conclusions or interpretations. The abstract should be an objective representation of the article: it must not contain results which are not presented and substantiated in the main text and should not exaggerate the main conclusions.

My abstract, as now written, fits these guidelines. Specifically,

Background: Using the same data set, a researcher can obtain very different reduced form estimates just by assuming different macroeconomic models.

Methods: Reiterative Truncated Projected Least Squares (RTPLS) or Variable Slope Generalized Least Squares (VSGLS) can be used to estimate total derivatives that are not model dependent.  These estimates can be used to pick between competing macro models, improve current models, or create new models.

Results: A selected survey of RTPLS estimates in the literature reveals several common patterns: (1) as income inequality has surged around the world, the effect of changes in government spending (G), exports (X), and money supply (M-1) on Gross Domestic Product (GDP) have plummeted, (2) decreases in G, X, and M-1 cause GDP to fall more than equal increases in G, X, and M-1 cause GDP to rise, and (3) unusually large increases in G and M-1 cause their effect on GDP to plummet.

Conclusion: These common patterns fit with a global glut of savings hypothesis, which predicts that an increase in savings will not cause an increase in production expanding investment. An appropriate model could be built around the idea that investors have a choice between investing to increase production or investing to earn rent or interest.

Furthermore, as instructed, I have not included these headings, and my abstract is exactly 200 words

  1. Introduction: Secondly, I revise the same issues in the Introduction, which will need great work to be improved. In particular, I suggest adopting a commonly used scheme, structuring this section as follows:
  •      Broad theme or topic;
    •       Academic and practical importance;
    •       Literature summary;
    •       Gaps, inconsistencies, controversies to be addressed;
    •       Focus of the study and research question(s), here or in Section 2;
    •       Applied methods;
    •       Main results and contributions;
    •       Structure of the article (remainder).
    At the moment, some of these parts are a little bit vague (i.e., theoretical and practical relevance, research gap) or missing (i.e., paper’s main results and implications, remainder). Moreover, you should try to better position your paper into the current scientific debate. In this sense, take particular care also to avoid supporting recent trends with old references. This happens even in the first lines of the article and several other times throughout the paper (i.e., also with references published in 2010). I also suggest providing a brief explanation of what is variable slope and how developed the macro model since the Introduction to help international readers contextualizing your analysis and generalizing its results.

            As now written, my introduction divides into your categories as follows:

Broad Theme or Topic: Cogan et al. (2010) dramatically shows that a researcher can obtain very different estimates just by changing the macroeconomic model employed.  Specifically, Cogan et al. (2010) contrast the government spending multipliers derived from a Romer-Bernstein Keynesian model with a Taylor Keynesian model using the same data. They find that the Taylor model predicts a government spending multiplier of 1.4 in the first quarter, which then declines to zero in the 24th quarter of their data.  In contrast the government spending multiplier estimated using the Romer-Bernstein Keynesian model is approximately one in the first quarter and steadily rises to 1.6 by the 8th quarter where it stays until the 16th quarter. They then find even a third set of results using a Smets-Wouters Keynesian model. Which model is closest to the truth, and upon which should we build government policy?

Academic and practical importance: Often economists use models that are “hard wired” to produce the results that they want to find in order to “prove” the viewpoints that they want to prove. Too often economists gather around different campfires – one for classicalists, another for Keynesians, another for monetarists, one for rational expectations, another for real business cycle theorists, etc. – where we reinforce those around our own campfire with no accepted way of picking which campfire is best.  What is needed are empirical tests that are not model dependent which can give guidance to model builders on what models would fit reality the closest.

Literature Summary (Note: the literature given here is what is necessary for an introduction, additional literature surveys occur later in the paper): Fortunately, in a recent Journal of Risk and Financial Management article, Leightner, Inoue, and Lafaye de Micheaux (2021) explain how total derivatives between an independent and dependent variable can be estimated that capture the effects of omitted variables without having to model all the possible ways that the two variables may interact.  The key intuition that underlies their methods is that the combined influence of all omitted variables determines the relative vertical position of observations. Thus, that relative vertical position can be used to capture the influence of omitted variables.  Leightner, Inoue, and Lafaye de Micheaux’s methods produce a separate slope estimate for every observation where the differences in these slope estimates are due to omitted variables. An important advantage of their methods is that they are not model dependent.  For example, their methods can be used to estimate a total derivative between government spending (G) and gross domestic product (GDP) without having to assume either a classical model (that has built-in mechanisms to cause dGDP/dG to equal zero) or a Keynesian model (that has built-in mechanisms to cause dGDP/dG to be greater than one).  Thus their methods can be used to estimate key total derivatives, which can then be used to pick between competing models or, even better, to create new models that are more realistic than any of our current choices.

Gaps, inconsistencies, controversies to be addressed and Focus of the study and research questions; For this paper, these are addressed in the “Academic and Practical Importance” part stated above.

Applied Methods are introduced in the Literature summary given above and expanded upon when I say: Several points need to be emphasized. First, variable slope estimation methods are not a substitute for macroeconomic modelling; variable slope estimation and macroeconomic models are complements that, when used together, can help policy makers make better choices. One major disadvantage of variable slope estimation is that they cannot tell the mechanisms that underlie the estimated total derivatives; we need macroeconomic models to understand these mechanisms.  Understanding the mechanisms that underlie the estimated total derivatives is crucial to the creation of optimal economic policies.

               Second, the total derivative estimates produced by variable slope estimation methods capture all the ways that the independent and dependent variables are related.  For example, if a government coordinates fiscal and monetary policy – cutting both spending and the money supply to combat inflation and increasing both to combat unemployment – then the total derivatives produced by variable slope estimation will capture that coordination. In other words, the variable slope fiscal spending multiplier estimate (dGDP/dG) does not hold the money supply constant if fiscal and monetary policy reinforce each other. Likewise, if a government increases spending to combat unemployment and that country’s monetary authorities simultaneously cut the money supply out of fear of inflation, then the dGDP/dG estimates will be diminished by the offsetting effects of contrary monetary policy. 

               Third, if government spending in year t is correlated with government spending in year t-1 and government spending in t-1 continues to effect GDP in year t, then variable slope estimates of dGDP/dG in year t will include the lingering effects of government spending in t-1.  This lingering effect can be diminished by first differencing the data before conducting the analysis. A comparison of the dGDP/dG estimates without first differencing to dGDP/dG estimates after first differencing could give a researcher information about the strength of lingering effects. Furthermore, if the data is first differenced, then variable slope estimation could be used to estimate dGDPt/dGt, dGDPt/dGt-1, dGDPt/dGt-2, etc. An alternative way of gaining dynamic information from variable slope estimations is based upon changing the time period of analysis. For example, Leightner and Inoue (2008) explored the effects of China’s exchange rate on the exchange rates of other Asian countries varying the time period of analysis from monthly to quarterly to annually.

               Fourth, even without first differencing or adjusting the time period of analysis, variable slope estimates contain dynamic information because they produce a separate slope estimate for every observation. For example, East and West Germany were reunited in 1990 at which time German government spending rose noticeably. Leightner (2015) used a variable slope estimation method to estimate dGDP/dG for Germany.  He found that dGDP/dG rose from 4.81 in the first quarter of 1988 to 5.19 in the fourth quarter of 1989 and then fell to 4.89 by the third quarter of 1992. The rise in dGDP/dG between 1988 and 1989 could be due to rising private expectations leading up to the reunification, and the fall in dGDP/dG fits what he found for many countries – unusually large increases in fiscal or monetary policy almost always result in a noticeable fall in the effectiveness of those policies.

This article will briefly explain the methods used by Leightner, Inoue, and Lafaye de Micheaux (2021).  One of their methods (reiterative truncated projected least squares – RTPLS) has been used to estimate many of the key relationships embedded in macroeconomic models.  A selected survey of these estimates will be presented and their implications for macroeconomic modelling explained.  This paper also presents evidence that there is a global glut of savings (where an increase in savings is not associated with an increase in production expanding investment) and challenges macroeconomic model builders to model this global glut of savings and its implications.

Main Results and Contributions and Structure of the article (reminder): The remainder of this paper is structured as follows.  Section 2 explains variable slope estimation methods and does not contribute significant new information to the literature. Section 3 surveys (a) selected variable slope estimates in the literature and (b) published evidence that there is a global glut of savings.  Section 3 contributes to the literature by (a) showing how the variable slope estimates in the literature fit with a global glut of savings, (b) making suggestions on how existing macroeconomic models can be improved based on these published variable slope estimates, and (c) suggesting an equation that could form the backbone of an entirely new macroeconomic model centered on the global glut of savings.  Section 4 concludes. 

3.Please add a literature review part to identify the methods and variables. Theoretical discussion and later implications of theory in research are very crucial. Furthermore, I am not biassed in your introduction. , considering Section 2, I suggest mobilising part of those concepts in the other paragraphs of the literature review since I cannot find any great value added by this section of the introduction.  

            I survey the RTPLS methods literature on lines 120-256.

            I survey the RTPLS estimate literature on lines 257-261 and 406-480

I discuss differences between the classical and Keynesian models on lines 265-283, 288-301, and 375-387.

I discuss global glut of savings models on lines 284-291.

I survey the evidence for a global glut of savings on lines 302-347.

I survey the New Keynesian models on lines 393-405

I admit that Section 2 does not noticeably add to the literature in the last paragraph of the introduction.

  1. Please check the equations of Methods part. I need justification more as why did you choose the method. 

The reasons I use these methods are given in the first three paragraphs of the paper and the first paragraph of the conclusion – past researchers have shown that the model assumed drastically changes the empirical results found even when the same data is used.  Variable slope estimation methods are not model dependent and the estimates they produce can be used to pick between competing models and to improve existing models. 

I have checked the equations in the Methods section many times – they are correct.

  1. Result part is ok.

            Thank you.  The results section contains the paper’s major contributions to the literature.

  1. Fifth, I suggest also giving more attention to developing the conclusion of your results, trying to better link it to the current scientific debate and the literature leveraged in Section 2 and 3. Reading this paragraph, I still cannot capture the distinctive elements that support the relevance of your study. Similarly, the theoretical and practical implications of your work, as well as its limitations and possible further developments, should be added in the Conclusion paragraph or in a separate section of the paper. Some great work is required in this regard.

The relevance is that which model is picked drastically affects the empirical results.  Variable slope estimation provides a very good way to find empirical evidence to guide a researcher in his or her choice of models.  I know of no better way to pick between competing macroeconomic models.  This is stated in the beginning paragraphs of the paper and of the conclusion.

To the conclusion, I have added, One major advantage of RTPS and VSGLS estimates is that you can see how the relationship between the independent variable and the dependent variables have changed over time due to omitted variables. Another major advantage is data is only needed on the independent and dependent variables – a researcher does not have to develop an entire macroeconomic model and estimate every equation in that model in order to derive a total derivative.  However, one disadvantage of RTPLS and VSGLS is their estimates do not tell the researcher the mechanisms through with the estimated relationship functions. RTPLS and VSGLS estimates can be used to pick which of competing models best fits reality.  

Thank you 

Good Luck with your research.

Round 2

Reviewer 2 Report

The authors have provided comments to all the issues that were raised. I think that their answer is reasonable given the limited time for revision.

Author Response

The referee said, "The authors have provided comments to all the issues that were raised. I think that their answer is reasonable given the limited time for revision."

My response: Thank you very much.  

Reviewer 3 Report

Still there is a vague in the revised MS by the authors.

1. Add more latest References. 

2. Why did you choose this method as objective-need for clarification with justified theories? 

Good luck!

Author Response

The referee made two requests: (1) Add more latest References and (2) Why did you choose this method as objective-need for clarification with justified theories? 

Concerning point 1, I am puzzled.  In the referee's first set of comments, the referee specifically mentioned that the reference dated 2010 was too old.  Since I had references dated 1935, 2000, 2005, and 2008, the referee's focus on 2010 makes me believe that he or she had a particular problem with my reference to Cogan et al. (2010).  Thus in response I have added the first footnote which says, "Cogen et al. (2010) provide the clearest and most dramatic evidence that picking a model can determine the empirical results produced.  Their article was seminal – as of June 2022 it has been cited 938 times. I know of no recent publications that reproduce their results nor do I know of papers that repudiate their results. It inspired this paper."

Concerning the timeliness of my references.  I have 26 references, ten of which are dated 2021 or 2022.  Furthermore 21 of my 26 references are dated 2012 or later.  If the referee and editor want me to add "more latest references", please tell me how recent these added references must be and how many more to add.

Concerning point 2, I added to lines 104-106, "The reason these methods are used is that they do not assume any model, and thus produce estimates that can be used to pick which model is best and to improve current models."